# Unlocking the Potential of Knowledge Distillation: The Role of Teacher Calibration

## Abstract

Knowledge distillation (KD) is one of the successful deep learning compression methods for edge devices, transferring the knowledge from a large model, known as the *teacher*, to a smaller model, referred to as the *student*. KD has demonstrated remarkable performance since its first introduction. However, recent research in KD reveals that using a higher-performance teacher network does not guarantee better performance of the student network. This naturally leads to a question about the criterion for choosing an appropriate teacher. In this paper, we reveal that there is a strong correlation between the calibration error of the teacher and the accuracy of the student. Therefore, we claim that the calibration error of the teacher model can be a selection criterion for knowledge distillation. Furthermore, we demonstrate that the performance of KD can be improved by simply applying a temperature-based calibration method that reduces the teacher's calibration error. Our algorithm can be easily applied to other methods, and when applied on top of the current state-of-the-art (SOTA) model, it achieves a new SOTA performance.

## 1 Introduction

In deep neural networks (DNNs), the most popular methods of model compression techniques are network quantization, network pruning, and knowledge distillation. Among them, knowledge distillation (KD) is a model training strategy that boosts the performance of a smaller DNN, so that it can replace a larger DNN. The goal of KD is to successfully transfer knowledge from a larger network, known as the teacher, to a smaller network, called the student. This process naturally yields two design criteria of KD: First, the choice of the distillation method used to transfer the knowledge of the teacher to the student, and second, the choice of a specific teacher for the student network[1].

The majority of contemporary works explore state-of-the-art (SOTA) distillation methods using fixed sets of teacher-student pairs. However, studying the latter criterion of choosing an appropriate teacher is important as well, as the simultaneous consideration of the two design criteria can ultimately provide a more optimized solution. For example, recent studies (Cho & Hariharan, 2019; Zong et al., 2022; Beyer et al., 2022) show that a better teacher does not always guarantee better students in knowledge distillation, and these findings highlight the need for methods of assessing a teacher and identifying a 'good teacher'. This paper deals with this issue. By analyzing the learning dynamics of deep learning empirically, we provide a high-level explanation about the question "which models induce effective KD". As an answer, we demonstrate that the calibration error of the teacher plays an important role, and significant performance improvements can be achieved in the KD process by simply applying the calibration method to the teacher network. We found that not only the standard KD but also the highly fine-tuned state-of-the-art KD method that has been designed in a sophisticated manner can benefit from the calibration method.

To delve into the details of our proposed method, we first propose using the calibration error as a criterion to evaluate the performance of the teacher. In statistics, calibration refers to the process of adjusting a probability model to ensure that its predictions closely align with actual probability. For instance, if the model predicts output with a probability (confidence) of 70% for a set of particular data samples, then ideally, 70% of those predictions should be correct to be considered well-calibrated. In contrast, a poorly calibrated model might predict an outcome with, for example, a

---

[1]In many real-world applications, a student network is preselected to meet the deployment conditions.

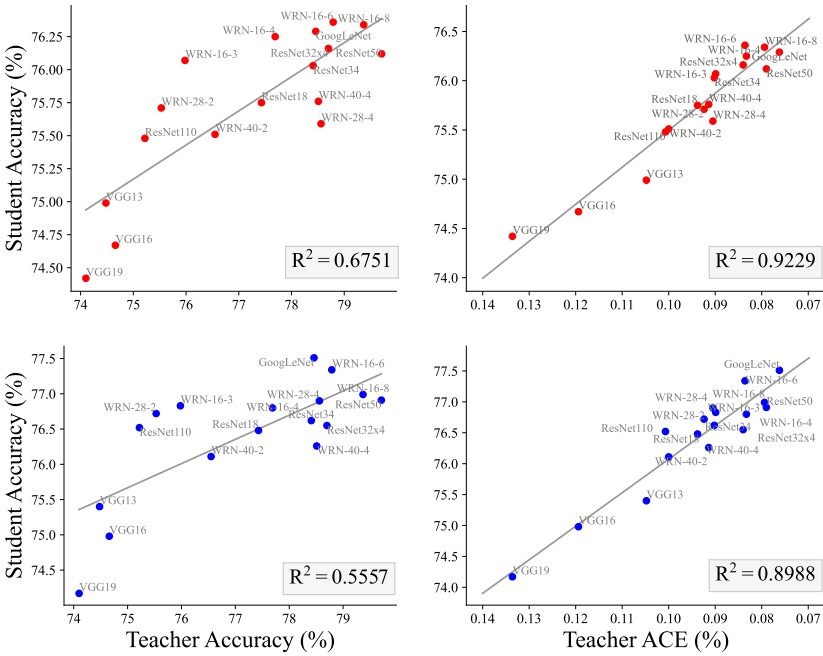

Figure 1: **Correlation for Teacher Accuracy and Student Accuracy (left) and Teacher Adaptive Calibration Error(ACE) and Student Accuracy (right).** The figure presents the outcomes of Knowledge Distillation (KD) training using 17 distinct pretrained teacher models. Each point indicates the teacher used for KD. Two student architectures are examined: WRN-16-2 (top) and ShuffleNetV2 (bottom). All experiments are conducted on the CIFAR-100 dataset.

99% probability, even though only 70% of those predictions are accurate. Poorly calibrated models like this case can be problematic in real-world applications. For example, in medical diagnostics or autonomous vehicle decisions, misplaced confidence can lead to severe consequences. Therefore, reducing calibration error is crucial for these applications.

When it comes to deep learning models, Guo et al. (2017) raised the issue that calibration error tends to increase as models become increasingly complex and achieve higher performance. As a result, follow-up studies have actively analyzed characteristics of calibration error in deep learning models (Nixon et al., 2019). In our research, we demonstrate that a well-calibrated model is not only probabilistically reliable but also effective when used as a teacher in knowledge distillation.

As observed in Fig. 1, there is a tendency for the teacher's calibration error to be low when the student's performance is high in the context of KD. Based on these findings, we propose that the calibration error of the teacher model can serve as a better criterion for selecting an effective teacher when employing knowledge distillation. Furthermore, this paper demonstrates that reducing the teacher model's calibration error can improve the performance of KD. By applying a simple temperature-based calibration method to teachers, we achieve consistent performance improvement against state-of-the-art models. Our experiments provide empirical evidence that the calibration error plays a crucial role in KD.

In summary, the main contributions of this paper are as follows:

- We experimentally show that the calibration error of the teacher negatively correlates with the performance of the student network in the process of KD. This finding enables the use of calibration error as a criterion for selecting an effective teacher.

- We demonstrate that the performance of knowledge distillation can be significantly improved by merely applying a simple calibration method to the teacher model. With the application of this simple technique to the previous SOTA method, we could establish new SOTAs in various experimental settings.

## 2 RELATED WORK

### 2.1 KNOWLEDGE DISTILLATION

Knowledge Distillation (KD) is one of the deep learning compression techniques introduced by Hinton et al. (2015). It leverages information from a larger, high-performing teacher model to train a smaller student model. By training students using KD, students can achieve better performance without additional memory and computational complexity.

Most papers of KD proposed techniques to boost the performance of the student. FitNet (Romero et al., 2014) and FT (Kim et al., 2018) utilized auxiliary networks called 'regressor' to assist the process of feature-map distillation. Methods such as DML (Zhang et al., 2018) and AFD (Chung et al., 2020) proposed online distillation methods that train the teacher and student simultaneously. TAKD (Mirzadeh et al., 2020) and DGKD (Son et al., 2021) proposed utilizing 'teacher assistants' that help bridge the gap between the student and the teacher. Recently, methods such as MLLD (Jin et al., 2023) and ReviewKD (Chen et al., 2021) set SOTA performance in logit distillation and feature distillation, respectively.

However, with numerous distillation methods that try to achieve so-called SOTA performance, the reason why each method should perform better than previous literature still remains very obscure. Thus, many studies discussed the fundamental reasons for the performance enhancements of KD. Furlanello et al. (2018) (BAN), and Zhao et al. (2022) (DKD) analyzed the effect of label smoothing and negative logits on KD. Cho & Hariharan (2019) showed that the better teacher did not promise the better performance of the student network and provided a solution based on early-stopping. IPWD (Niu et al., 2022) showed the importance of bridging the classwise knowledge imbalance gap between the teacher and the student. Menon et al. (2021); Zhou et al. (2021) speculated the role of KD based on the bias-variance trade-offs. Following the findings of Cho & Hariharan (2019), in this paper, we inspect the role of the teacher's 'calibration error' on knowledge distillation.

### 2.2 DEEP LEARNING CALIBRATION

Model calibration refers to making the model's actual accuracy reflect the actual confidence or vice-versa. Since the machine learning era, lots of studies have been conducted to decrease the calibration error of a model. Some representative examples are Histogram binning (Zadrozny & Elkan, 2001), Isotonic regression(Zadrozny & Elkan, 2002), Platt scaling(Platt et al., 1999), vector scaling, and temperature scaling. Rather recently, Guo et al. (2017) reported that simply using temperature scaling calibrates well enough for deep learning models. Following the work of Guo et al. (2017), we investigate the usage of temperature scaling on the DNNs, which yields no additional costs in training or testing.

## 3 CHOOSING A BETTER TEACHER BASED ON CALIBRATION ERROR

### 3.1 MEASURING CALIBRATION ERROR FOR MODERN DEEP LEARNING MODELS

Expected Calibration Error (ECE) is the metric for measuring the calibration error of a certain probabilistic model. The ECE value can be acquired as follows:

$$\text{ECE} = \sum_{m=1}^{M} \frac{|B_m|}{n} \left| \text{acc}(B_m) - \text{conf}(B_m) \right|, \tag{1}$$

where $M$ is the number of bins dividing the accuracy space of [0,1] evenly, $B_m$ is the set of samples belonging to interval $m$, $\text{acc}(B_m)$ refers to the accuracy of interval $m$, and $\text{conf}(B_m)$ refers to the average prediction probability (confidence) of samples in interval $m$.

Originally, ECE was designed for binary classification (DeGroot & Fienberg, 1983) and has been adapted for multi-class problems by treating the problem with $K$ classes as a set of binary predictions: one for the target label and the other for non-target labels. However, this approach neglects the information that deep learning models may capture between non-target classes. Such ignorance is problematic for knowledge distillation, which inherently involves transferring information about inter-class relationships (Furlanello et al., 2018). Additionally, ECE suffers from another limitation

in that it creates bins by evenly spacing the probability space. This trait results in scenarios where fewer samples at lower probability levels contribute significantly to the ECE. Thus, ECE may not be the most effective metric for measuring the calibration of deep learning models that often display highly overconfident probability distributions.

Given these limitations, this paper employs Adaptive Calibration Error (ACE) (Nixon et al., 2019) as a metric for measuring calibration error. ACE accounts for multi-class predictions and employs adaptive binning to equalize the number of samples in each bin. The formula for ACE is as follows:

$$\text{ACE} = \frac{1}{KR} \sum_{k=1}^{K} \sum_{r=1}^{R} |\text{acc}(r,k) - \text{conf}(r,k)|. \tag{2}$$

Here, $K$ represents the number of classes, and $R$ signifies the number of bins. The terms $\text{acc}(r,k)$ and $\text{conf}(r,k)$ refer to the accuracy and confidence (output probability), respectively, of the samples belonging to class $k$ in the $r$-th bin. The calibration range $r$ is defined by the $\lfloor N/R \rfloor$th index, where $N$ is the number of samples. ACE offers several advantages over ECE. Specifically, it computes errors for non-target probabilities and allocates an equal number of samples to each bin. These features make it especially well-suited for assessing the calibration of deep learning models when the probability distribution is skewed.

Additionally, to identify the source of a model's calibration error, we can decompose the ECE into its components: overconfident ECE ($\text{ECE}_{over}$) and underconfident ECE ($\text{ECE}_{under}$). The decomposed ECE can be attained as follows:

$$\text{ECE}_{over} = \sum_{m=1}^{M} \frac{|B_m|}{n} \max(\text{conf}(B_m) - \text{acc}(B_m), 0), \tag{3}$$

$$\text{ECE}_{under} = \sum_{m=1}^{M} \frac{|B_m|}{n} \max(\text{acc}(B_m) - \text{conf}(B_m), 0). \tag{4}$$

$\text{ECE}_{over}$ represents the sum of the overconfident components of the calibration error, while $\text{ECE}_{under}$ captures the sum of the underconfident components. These two metrics allow us to determine whether the calibration error of a model predominantly originates from its overconfident or underconfident predictions. Our empirical evaluation across a variety of models confirms that calibration errors of teachers primarily arise from overconfident predictions when models are trained using standard procedures. For more detailed insights into the calibration errors of the various teacher models used in this study, please refer to Section A in the appendix.

## 3.2 Teacher model selection by measuring calibration error

In section 1, we highlighted that the accuracy of the teacher is not the best indicator for the better accuracy of the student model, proposing the calibration error as a more reliable metric. To validate this hypothesis, we examined the correlation between the accuracy of the student model and both the accuracy and calibration error of the teacher model. Figure 1 clearly illustrates a strong correlation between the calibration error of the teacher and the accuracy of the student models. The $R^2$ values for the correlation between the teacher model's calibration error and the student model's accuracy are 0.9229 for WRN-16-2 and 0.8998 for ShuffleNetV2, respectively. The accuracies of the teacher models show relatively lower correlations with the student's accuracy, where the values are only 0.6751 and 0.5557.

These tendencies suggest that teachers with lower calibration errors perform better in Knowledge Distillation (KD). Compared to teachers with higher calibration errors, those with lower errors have two comparative advantages. First, they provide a basis for KL divergence to take effect in KD. KL divergence fundamentally measures the gap between probability distributions, and in this context, a lower calibration error allows the teacher's output to form a more accurate probability distribution. Essentially, calibration error measures how well the model's output probability reflects accuracy; hence, a lower calibration error means the model forms a better mathematical probability distribution with respect to the input distribution. Second, teachers with lower calibration errors act as stronger regularizers. The ground truth label can be considered as a label with 100% confidence, and teachers

Table 1: **Comparison of Teacher Models which has similar accuracy.** Evaluation of the KD Accuracy of student models WRN-16-2 and ShuffleNetV2 with WRN-40-4, and GoogLeNet teacher on the CIFAR100 dataset. Bold values denote superior performance in each category.

| | Teacher | | | Student KD Accuracy(%) | |
|---|---|---|---|---|---|
| Model | Accuracy(%) | ACE(%) | | WRN-16-2 | ShuffleNetV2 |
| WRN-40-4 | **78.51** | 0.0914 | | 75.76 | 76.26 |
| GoogLeNet | 78.46 | **0.0762** | | **76.29** | **76.55** |

with lower calibration errors (i.e., less overconfident teachers) play a larger role as label smoothers for the true label. This amplifies one of the benefits of KD, which takes its role as a regularizer.

Benefits become immediately apparent when ACE is utilized for selecting the teacher model. Referring to Table. 1 allows for a comparison of teacher accuracy between GoogLeNet and WRN-40-4 models. Although WRN boasts higher accuracy, better results are obtainable from GoogLeNet when examining the actual learning outcomes of the student models. Without using ACE, predicting such results would be difficult. Generally, if judgments were made based on the similarity of architecture or the accuracy of the teacher model, it would lead to the choosing of WRN-40-4 as a teacher, which results in lower student accuracy.

## 4    MAKING A BETTER TEACHER VIA CALIBRATION METHOD

To investigate the effect of calibration on KD, it is essential to focus on how the teachers function in the process of KD. KD does not directly transfer the parameters of a model but applies the input data to the teacher model to abstract the teacher's information and transmit it to the student. Therefore, what we should focus on is not the parameters of the model itself but the probability simplex to which each data point is mapped via the model. By mimicking this mapping, the student can acquire not only the true class of a given input but also information about the relative relationships and uncertainties among classes.

However, there arises a question of whether this process can truly be performed successfully. Typically, cross-entropy (CE) or its variations are used for training the model, and one of the characteristics of these losses is that these losses can never reach zero, pushing the value of the highest-valued logit larger and larger, thus producing overconfident outputs (Soudry et al., 2022).

The overconfident outputs may stymie the knowledge distillation process for the following reasons. Because the KD loss (typically KL (Kullback-Leibler) divergence) is applied on top of the conventional CE loss, which utilizes the ground truth one-hot vector as the target, if the teacher network outputs with high confidence, the probability vector (softmax output) will be very close to the ground truth one-hot vector and the KD does not take significant effect on the learning process. Furthermore, when the teacher network outputs a wrong class as its prediction with high confidence, the KD loss competes with the ground truth CE loss, preventing successful learning. Therefore, it is imperative to find non-overconfident teachers for successful KD.

In the previous section, we explored the possibility of using calibration error as a criterion for teacher selection in KD. However, we argue that the calibration error has more potential beyond. We have discussed the overconfidence prior present in deep learning models. Regarding calibration error functions as a metric to assess the alignment between a model's probability (confidence) distribution and the accuracy distribution, this prior contributes to producing a high calibration error. If we can address this issue, we can anticipate an improvement in the model's calibration error.

The overconfidence issue arises when the logit value of the predicted class is excessively high while the remaining logit values are substantially lower. We resolve this issue with a straightforward yet effective solution: by incorporating temperature scaling in the softmax calculation phase, we aim to smoothen the probability distribution. As a result of this adjustment, the confidence distribution of the teacher model becomes smoother, thereby alleviating the overconfidence issue. In this section, we demonstrate that this strategy can significantly enhance the baseline performance. Furthermore, this method proves to be effective not only with simplistic KD approaches but also with finely tuned state-of-the-art KD techniques.

## 4.1 IMPLEMENTATION DETAIL

There are many calibration methods such as Platt scaling (Platt et al., 1999) and isotonic regression (Zadrozny & Elkan, 2002), matrix scaling, vector scaling, and temperature scaling (Guo et al., 2017). We utilized temperature scaling, which only has one scalar hyperparameter $T$. Thus, it does not need an independent validation set for training while still reducing the overconfident calibration error effectively. Eq. 5 below shows the temperature scaling formulation:

$$\hat{p}_i = \frac{e^{z_i/T}}{\sum_j^K e^{z_j/T}},$$ (5)

where $K$ represents the number of classes and $T$ is the temperature parameter, and $z \in \mathbb{R}^K$ denotes the output logit vector of the model, whose $i$-th element being denoted as $z_i$. Setting $T = 1$ makes the equation equivalent to the standard softmax function, and when $T > 1$, it softens the output probability. Importantly, changing the temperature does not affect the order of the output probability, meaning that it does not influence the model's prediction accuracy. Thus, changing the temperature gives a way of changing the model's confidence for a fixed accuracy. Unless otherwise specified in the subsequent experiments, we set the default hyperparameter $T = 1.5$ and discuss the effect of varying $T$ in Section 4.2.

We should note that temperature scaling is also commonly used in KD to distill non-target label information of teachers. However, in the context of KD, temperature scaling is typically applied to both the teacher and the student logits. The purpose of temperature scaling in conventional KD is to make the student's logit follow the teacher's logit effectively even for non-target labels. In contrast, the purpose of temperature scaling in our calibration is to enable the student to learn from a well-calibrated teacher. In our experiments, we utilize both temperature scaling techniques. We apply temperature scaling solely to the teacher to reduce the overconfident calibration error and also apply temperature scaling to the teacher and student for better non-target logit distillation. For example, if the temperature scaling value of 2 is used for both the teacher and student model in the original method, our method scales the teacher's temperature further by 1.5 times resulting in a temperature of 3 for the teacher, while the student's temperature remains at 2. To the best of our knowledge, this work is the first approach to apply the calibration method specifically to KD.

After applying temperature scaling to the pre-trained teacher model, the remaining training steps follow the general knowledge distillation process as described by (Hinton et al., 2015), which uses the cross-entropy loss with the true label and the KL divergence loss with the teacher output. Further training details can be found in Section B on the appendix.

## 4.2 CALIBRATION METHOD CAN IMPROVE KD

In our experiments, we investigated the potential of calibration methods that enhance the performance of knowledge distillation. Our findings demonstrate that a teacher model calibrated through temperature scaling improves KD performance. Our experiment also shows higher KD performance when operated in a slightly underconfident state through elevated temperature settings. Figure 2 provides an insight into how the performance of KD varies with the changing temperature of the teacher model. The graph illustrates that when an appropriate temperature (T) is set, the overconfident ECE of the teacher model is reduced. Additionally, our findings indicate that the teacher model demonstrates superior performance even when it is somewhat underconfident as a result of higher temperature settings. This can be attributed to the fact that the true labels act as highly overconfident labels during KD, and consequently, employing an underconfident teacher model in conjunction with these true labels can yield more balanced probabilities. Based on these insights, we chose a temperature setting of T=1.5, aiming to reduce the overconfident tendencies in the teacher model's probability estimates while also preventing it from becoming excessively underconfident. This approach thereby improves the overall performance of KD.

We conducted experiments on Table 2 using 17 different teacher models. The experiment aimed to test whether mitigating overconfident calibration errors through temperature scaling could consistently enhance performance across various architectures. We compared the validation accuracy of student models using temperature-scaled KD, which we refer to as "Ours," against those using standard KD. As evidenced in Table 2, performance improvement was observed in the majority of

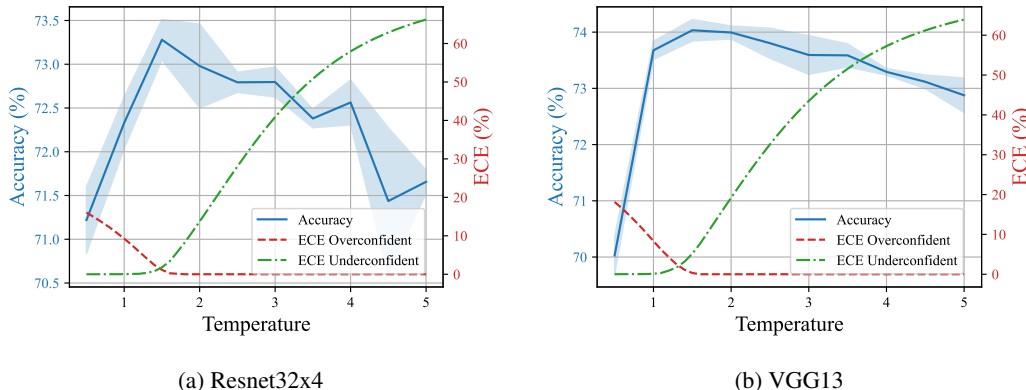

| | (a) Resnet32x4 | (b) VGG13 |

Figure 2: **Effect of temperature of teacher.** The figure illustrates the impact of varying the temperature parameter on the performance of student models. Two different teacher-student architectures, (a) Teacher: ResNet32x4, Student: WRN-16-2, and (b) Teacher: VGG13, Student: VGG8, are considered. Experiments are conducted on the CIFAR-100 dataset. Each experimental setting was run three times to calculate the mean and standard deviation.

Table 2: **KD results with temperature scaled teachers.** The table shows the teacher model's accuracy and the student model's accuracy under standard KD and temperature-scaled KD. ResNet56 is used for the student model which has an original accuracy of 73.73%.

| Teacher architecture | vgg19 | vgg13 | vgg16 | resnet110 | wrn-28-2 | wrn-16-3 | wrn-40-2 | ResNet18 | wrn-16-4 |
|---|---|---|---|---|---|---|---|---|---|
| Teacher accuracy | 74.10 | 74.48 | 74.66 | 75.22 | 75.53 | 75.98 | 76.55 | 77.43 | 77.69 |
| KD | 72.90 | 73.99 | 73.39 | 74.98 | 75.40 | 75.37 | 74.90 | 74.36 | 75.33 |
| KD + Ours | **73.37** | **74.33** | **73.90** | **75.39** | **76.12** | **75.95** | **75.20** | **74.68** | **75.95** |

| Teacher architecture | ResNet34 | GoogleNet | wrn-40-4 | wrn-28-4 | resnet32x4 | wrn-16-6 | wrn-16-8 | ResNet50 |
|---|---|---|---|---|---|---|---|---|
| Teacher accuracy | 78.41 | 78.46 | 78.51 | 78.56 | 78.70 | 78.79 | 79.37 | 79.71 |
| KD | 74.53 | 74.81 | 75.00 | 74.27 | 74.95 | 75.55 | 75.16 | 75.13 |
| KD + Ours | **74.8** | **75.6** | **75.12** | **74.53** | **75.23** | **75.62** | **75.23** | **75.48** |

the models tested. These findings affirm that the use of a temperature-scaled teacher model is not dependent on the specific teacher architecture and generally enhances performance.

## 4.3 COMPARISON WITH THE STATE-OF-THE-ART METHODS

In our study, we conducted a comprehensive evaluation of our proposed method against state-of-the-art (SOTA) models in the context of Knowledge Distillation (KD). The key innovation in our approach is the application of temperature scaling only to the logits of the teacher model before executing KD, which significantly enhances the performance of the student models. Our experiments were conducted on multiple datasets, including CIFAR-100 (Krizhevsky, 2012) and ImageNet (Russakovsky et al., 2015). The results of our experiments are presented in Tables 3, 4, 5, and 6 indicate notable performance enhancements when our temperature-scaling method with $T = 1.5$ is applied.

**Experiment on CIFAR-100** Table 3 reports the experiment results on the homogenous architectures, and Table 4 reports the experiment results on the heterogeneous architectures. With various choices of the network pairs, using temperature-scaled teachers shows consistent improvements in the student accuracies. It is notable that although it is quite simple, additionally applying the teacher temperature scaling to the previous SOTA method (MLLD + Ours) sets up the new SOTA results. This is quite impressive considering that the previous SOTA methods in the feature distillation (ReviewKD) and logit distillation (MLLD) introduce various kinds of auxiliary networks or additional loss terms. With these experiments, we verify that simply applying temperature scaling draws the better role of the teacher network. For some pairs, training the student with MLLD + Ours is the only way to surpass its corresponding teacher, whereas the other previous methods fail to do so. It

Table 3: **Results on the CIFAR-100 dataset with homogenous Teacher-Student architectures.** The table presents the experimental results of various state-of-the-art KD methods, including feature-based and logit-based distillation methods. Bold values highlight the top-performing method for each architecture. KD$^\dagger$ and MLLD$^\dagger$ represents the result implemented by ours.

| Method | | ResNet56 72.34 | ResNet110 74.31 | ResNet32x4 79.42 | WRN-40-2 75.61 | WRN-40-2 75.61 | VGG13 74.64 |
|---|---|---|---|---|---|---|---|
| | Teacher | ResNet56 72.34 | ResNet110 74.31 | ResNet32x4 79.42 | WRN-40-2 75.61 | WRN-40-2 75.61 | VGG13 74.64 |
| | Student | ResNet20 69.06 | ResNet32 71.14 | ResNet8x4 72.50 | WRN-16-2 73.26 | WRN-40-1 71.98 | VGG8 70.36 |
| Features | FitNet | 69.21 | 71.06 | 73.50 | 73.58 | 72.24 | 71.02 |
| | RKD | 69.61 | 71.82 | 71.90 | 73.35 | 72.22 | 71.48 |
| | CRD | 71.16 | 73.48 | 75.51 | 75.48 | 74.14 | 73.94 |
| | ReviewKD | 71.89 | 73.89 | 75.63 | 76.12 | 75.09 | 74.84 |
| Logits | KD$^\dagger$ | 70.90 | 73.62 | 75.69 | 75.33 | 73.43 | 73.96 |
| | KD + Ours | 71.23 | 74.19 | 76.01 | 75.85 | 74.46 | 74.23 |
| | DKD | 71.97 | 74.11 | 76.32 | 76.23 | 74.81 | 74.68 |
| | MLLD | 72.19 | 74.11 | 77.08 | 76.63 | 75.35 | 75.18 |
| | MLLD$^\dagger$ | 72.05 | 74.48 | 77.02 | 76.47 | 75.56 | 74.99 |
| | MLLD + Ours | **72.78** | **74.87** | **77.39** | **76.79** | **76.27** | **75.36** |

Table 4: **Results on the CIFAR-100 dataset with heterogenous Teacher-Student architectures.** The table presents the experimental results of various state-of-the-art KD methods, including feature-based and logit-based distillation methods. Bold values highlight the top-performing method for each architecture. KD$^\dagger$ and MLLD$^\dagger$ represents the result implemented by ours.

| Method | | ResNet32x4 79.42 | WRN-40-2 75.61 | VGG13 74.64 | ResNet50 79.34 | ResNet32x4 79.42 |
|---|---|---|---|---|---|---|
| | Teacher | ResNet32x4 79.42 | WRN-40-2 75.61 | VGG13 74.64 | ResNet50 79.34 | ResNet32x4 79.42 |
| | Student | ShuffleNet-V1 70.50 | ShuffleNet-V1 70.50 | MobileNet-V2 64.60 | MobileNet-V2 64.60 | ShuffleNet-V2 71.82 |
| Features | FitNet | 73.59 | 73.73 | 64.14 | 63.16 | 73.54 |
| | RKD | 72.28 | 72.21 | 64.52 | 64.43 | 73.21 |
| | CRD | 75.11 | 76.05 | 69.73 | 69.11 | 75.65 |
| | ReviewKD | 77.45 | 77.14 | 70.37 | 69.89 | 77.78 |
| Logits | KD$^\dagger$ | 72.69 | 73.13 | 65.07 | 64.78 | 76.15 |
| | KD + Ours | 73.66 | 73.88 | 68.6 | 67.27 | 75.98 |
| | DKD | 76.45 | 76.70 | 69.71 | 70.35 | 77.07 |
| | MLLD | 77.18 | 77.44 | 70.57 | 71.04 | 78.44 |
| | MLLD$^\dagger$ | 77.13 | 77.26 | 69.70 | 69.60 | 78.42 |
| | MLLD + Ours | **77.49** | **77.84** | **70.85** | **71.24** | **78.48** |

is valuable in the view of model compression in that the ultimate goal of knowledge distillation is to make the student able to replace the teacher.

**Experiment on ImageNet** In addition to the CIFAR-100 dataset, we extended our experiments to the ImageNet dataset, focusing on ResNet34-ResNet18 and ResNet50-MobileNetV1 architectures, as detailed in Table 5 and Table 6, respectively. Our approach demonstrates a marked improvement in performance relative to standard Knowledge Distillation (KD), consistent with our findings on the CIFAR-100 dataset. Furthermore, our method exhibits versatility by enhancing the performance of other techniques. For example, when integrated with the MLLD approach, our method achieves state-of-the-art (SOTA) performance in our implementation.

Additionally, when the MobileNet-V1 was used for the student in Table 6, the abnormal performance drop reveals the harmfulness of using the overconfident logits in the vanilla KD loss. Our method mitigates this by simply applying temperature to the teacher logit.

Table 5: Top-1 and top-5 accuracy (%) on the ImageNet validation. We set ResNet-34 as the teacher and ResNet-18 as the student. MLLD$^{\dagger}$ represents the result implemented by ours.

| | Method | | Features | | | | Logits | | | | | |
| | Teacher | Student | AT | OFD | CRD | RevieKD | KD | KD + Ours | DKD | MLLD | MLLD$^{\dagger}$ | MLLD + Ours |
|---|---|---|---|---|---|---|---|---|---|---|---|---|
| top-1 | 73.31 | 69.75 | 70.69 | 70.81 | 71.17 | 71.61 | 70.66 | 71.60 | 71.70 | **71.90** | 71.60 | 71.89 |
| top-5 | 91.42 | 89.07 | 90.01 | 89.98 | 90.13 | 90.51 | 89.88 | 90.24 | 90.41 | 90.55 | 90.68 | **90.69** |

Table 6: Top-1 and top-5 accuracy (%) on the ImageNet validation. We set ResNet-50 as the teacher and MobileNet-V1 as the student. MLLD$^{\dagger}$ represents the result implemented by ours.

| | Method | | Features | | | | Logits | | | | | |
| | Teacher | Student | AT | OFD | CRD | RevieKD | KD | KD + Ours | DKD | MLLD | MLLD$^{\dagger}$ | MLLD + Ours |
|---|---|---|---|---|---|---|---|---|---|---|---|---|
| top-1 | 76.16 | 68.87 | 69.56 | 71.25 | 71.37 | 72.56 | 68.58 | 71.55 | 72.05 | 73.01 | 73.05 | **73.09** |
| top-5 | 92.86 | 88.76 | 89.33 | 90.34 | 90.41 | 91.00 | 88.98 | 90.43 | 91.05 | 91.42 | 91.49 | **91.44** |

## 4.4 PROPERTY OF STUDENT TRAINED VIA CALIBRATED TEACHER

Table 7: **Properties of teacher and student models with calibration method.** The table presents the impact of temperature scaling on various metrics, including ECE, $ECE_{over}$, $ECE_{under}$ ACE, and ACC, for teachers and students. Bold values denote superior performance in each category.

| | Teacher: ResNet32x4 | | | | Student: ShuffleNetV1 | | | | |
| T | ECE | $ECE_{over}$ | $ECE_{under}$ | ACE | ECE | $ECE_{over}$ | $ECE_{under}$ | ACE | Student ACC |
|---|---|---|---|---|---|---|---|---|---|
| 1 | 9.3010 | 9.2661 | **0.0349** | **0.0806** | 16.8755 | 16.8755 | **0.0000** | 0.1299 | 72.69 |
| 1.5 | **2.9491** | **1.1564** | 1.7927 | 0.1310 | **8.7419** | **8.4847** | 0.2572 | **0.1006** | **73.66** |

| | Teacher: WRN-40-2 | | | | Student: WRN-16-2 | | | | |
| T | ECE | $ECE_{over}$ | $ECE_{under}$ | ACE | ECE | $ECE_{over}$ | $ECE_{under}$ | ACE | Student ACC |
|---|---|---|---|---|---|---|---|---|---|
| 1 | 11.2194 | 11.2003 | **0.0191** | 0.1000 | 11.1290 | 11.1251 | **0.0039** | 0.0948 | 75.33 |
| 1.5 | **3.0178** | **2.4437** | 0.5741 | **0.0892** | **3.5663** | **2.8599** | 0.7064 | **0.0885** | **75.85** |

Table 7 demonstrates the impact of using a teacher with a reduced calibration error on the student's calibration. When set to $T = 1.5$, we observe a significant reduction in the teacher's overconfident calibration error. Consequently, the student's accuracy improves, along with reductions in all the metrics of ECE, overconfident ECE, and ACE. Notably, in the case of the ResNet32x4-ShuffleNetV1 pair, although the teacher's ACE increases due to an increase in underconfident error, the student's ACE is actually reduced. This suggests that a slightly underconfident teacher, when combined with highly overconfident true labels, enables the student to learn the actual probabilities more effectively. These experimental results validate that our approach not only enhances the accuracy of the student model but also produces a student that is better calibrated. This implies that our approach can yield more reliable models, particularly beneficial when applied in real-world applications.

## 5 CONCLUSION

In this paper, we argued that the calibration error plays a crucial role in KD methodologies. this leads us to propose the calibration error of teachers as a new selection criterion. We have shown the validity of this criterion through various experiments. Furthermore, we discovered that by adjusting the temperature scaling to the model to reduce calibration error, substantial performance improvements can be achieved in standard KD methodologies. This approach is also applicable to existing state-of-the-art methods, demonstrating the possibility of additional performance enhancements through its application. Overall, our findings highlight the importance of the teacher calibration error in the knowledge distillation and providing a foundation for further advancements in KD methods

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

# A    PROPERTY OF PRETRAINED TEACHER MODELS

| | Teacher Properties | | | | | | Student KD Accuracy(%) | |
|---|---|---|---|---|---|---|---|---|
| **Model** | **Accuracy(%)** | **ECE (%)** | **$ECE_{over}$(%)** | **$ECE_{under}$(%)** | **ACE(%)** | **Params** | **WRN-16-2** | **ShuffleNetV2** |
| VGG19 | 74.10 | 16.4723 | 16.4723 | 0.0000 | 0.1336 | 20.1M | 74.42 | 74.17 |
| VGG13 | 74.48 | 8.4570 | 8.3338 | 0.1232 | 0.1048 | 9.5M | 74.99 | 75.40 |
| VGG16 | 74.66 | 12.8963 | 12.8946 | 0.0017 | 0.1194 | 14.8M | 74.67 | 74.98 |
| ResNet110 | 75.22 | 13.0475 | 13.0475 | 0.0000 | 0.1007 | 1.7M | 75.48 | 77.04 |
| WRN-28-2 | 75.53 | 7.1268 | 7.1268 | 0.0000 | 0.0924 | 1.5M | 75.71 | 76.72 |
| WRN-16-3 | 75.98 | 8.7114 | 8.6809 | 0.0305 | 0.0899 | 1.6M | 76.07 | 77.01 |
| WRN-40-2 | 76.55 | 11.2194 | 11.2003 | 0.0191 | 0.1000 | 2.3M | 75.51 | 76.11 |
| ResNet18 | 77.43 | 6.9784 | 6.8159 | 0.1625 | 0.0938 | 11.2M | 75.75 | 76.48 |
| WRN-16-4 | 77.69 | 7.5518 | 7.5394 | 0.0124 | 0.0833 | 2.8M | 76.25 | 76.8 |
| ResNet34 | 78.41 | 9.5015 | 9.4911 | 0.0104 | 0.0902 | 21.3M | 76.03 | 76.62 |
| GoogLeNet | 78.46 | 6.7732 | 6.7621 | 0.0111 | 0.0762 | 6.4M | 76.29 | 76.55 |
| WRN-40-4 | 78.51 | 10.3410 | 10.3410 | 0.0000 | 0.0914 | 9.0M | 75.76 | 76.26 |
| WRN-28-4 | 78.56 | 8.2353 | 8.2353 | 0.0000 | 0.0905 | 5.9M | 75.59 | 76.90 |
| ResNet32x4 | 78.70 | 9.3010 | 9.2661 | 0.0349 | 0.0840 | 7.4M | 76.16 | 76.55 |
| WRN-16-6 | 78.79 | 6.3329 | 6.3271 | 0.0058 | 0.0836 | 6.2M | 76.36 | 77.53 |
| WRN-16-8 | 79.37 | 5.6263 | 5.5865 | 0.0398 | 0.0794 | 11.0M | 76.34 | 76.99 |
| ResNet50 | 79.71 | 7.6640 | 7.6640 | 0.0000 | 0.0790 | 23.7M | 76.12 | 76.91 |

Table 8: **Properties of Teacher Models and Student Models used in this paper**: This table presents various metrics for teacher models and their corresponding student models accuracy trained with standard KD method on CIFAR-100 dataset.

Table 8 provides detailed properties of the teacher models that were used in Figure 1 in Section 3.2 to demonstrate the correlation between teacher ACE and student accuracy, and in Table 2 to show the enhanced KD performance achieved by temperature-scaled teachers across various models in Section 4.2. From this table, we can observe that teachers trained using conventional training schemes predominantly exhibit overconfident calibration errors, which justifies our approach to reducing the overconfident calibration error.

# B    IMPLEMENTATION DETAILS

## B.1    CIFAR-100

In our experiments, the training scheme for KD was configured based on a paper proposed by Zhao et al. (2022). We set the batch size to 128 and conducted training over a total of 240 epochs. The initial learning rate was configured at 0.5 and was decayed by a factor of 10 at epochs 150, 180, and 210. The SGD optimizer was used with a weight decay set to 5e-5 and a momentum of 0.9. The temperature parameter for KD was set to 4, and the loss function combined the cross-entropy with the true labels and the Kullback-Leibler divergence with the teacher's output with student output. The weights for these loss components were set at 0.1 and 0.9, respectively. The primary difference between the standard KD and our enhanced KD scheme lies in the application of temperature scaling to reduce overconfident calibration in the teacher model. For calibration purposes, a temperature of 1.5 was uniformly used across all KD + Ours experiments.

For the implementation of MLLD + Ours, we strictly followed the training scheme of MLLD as described in (Jin et al., 2023). However, we applied temperature scaling for calibration only when using the KL-divergence loss, one of the four types of losses (Cross entropy loss, KL-divergence loss, batch-level loss, class-level loss) employed in MLLD.

## B.2    IMAGENET

For KD training on ImageNet, we followed the training scheme proposed in Zhao et al. (2022). The batch size was set to 512, and the training was conducted over 100 epochs. The initial learning rate was set to 0.2 and was divided by 10 at epochs 30, 60, and 90. We used the SGD optimizer with a weight decay of 1e-4 and a momentum of 0.9. Similar to CIFAR-100, we employed cross-entropy

loss with the true labels and KL-divergence loss with the teacher output, assigning equal weights of 0.5 to each loss. The temperature for KD was set to 1. The only difference between KD and KD + Ours was the application of temperature scaling for calibrating the teacher model.

For the MLLD + Ours experiment, we adhered to the training scheme of MLLD as described in (Jin et al., 2023). We applied temperature scaling to the teacher's output for calibration, performed probability smoothing, and then computed the loss.

## C  LIMITATIONS AND FUTURE WORK.

**Limitations.** While this paper empirically demonstrates the significant role of overconfident calibration error in KD, it is important to note that our study is limited in scope to logit distillation methods. Feature distillation is also an important technique widely employed in deep learning applications. We have not explored the impact of calibration error on feature distillation methods, thereby indicating a need for further research in this area.

**Future Work.** The advancement of calibration methods holds considerable promise for their application in knowledge distillation. There are numerous opportunities for integrating more sophisticated calibration techniques into KD pipelines. Future work could involve evaluating the effectiveness of different calibration methods in the KD setting, potentially leading to more robust and accurate student models. Another avenue for research could be to understand the interplay between calibration and other aspects of KD, such as data augmentation, model complexity, and training dynamics. This could yield insights into developing an integrated framework for KD that accounts for both performance and calibration.

