# OpenReview forum: "Unlocking the Potential of Knowledge Distillation: The Role of Teacher Calibration"
_ICLR.cc/2024/Conference — Submitted to ICLR 2024_

### Official Review · Reviewer_vJXm · 2023-10-23

**Soundness:** 3 good
**Presentation:** 4 excellent
**Contribution:** 3 good
**Rating:** 6
**Confidence:** 4

**Summary:**

Contemporary studies in knowledge distillation have revealed that the better teacher models could not promised the better performance of the student models.
Motivated by this result, this paper aims to define the how to select the better teacher models.
To this end, the authors propose a simple yet effective calibration error for the metric.
This paper empirically demonstrates the student models well-learn the knowledge of the teacher models by applying the simple temperature scaling to the teacher models.

**Strengths:**

1. This paper reveals the calibration error is better than an accuracy for the metric on the knowledge distillation.
In addition, it is convincing using the adaptive calibration error (ACE) rather than the expected calibration error.
2. This paper not only covers the most of the related studies but also is easy to follow.

**Weaknesses:**

It is necessary to assess whether the proposed method works well even if various calibration studies are applied. And also they should be covered in the related works.

**Questions:**

1. In the introduction, the authors described that two things are important in the KD process. The first is to choose an effective distillation method, and the second is to choose the best one of teachers for students. But in real-world applications, under the deployment conditions, isn't it more typical to choose a student model that can learn the best teacher model?
2. Wouldn't it be possible to reverse when T=2 and T=4?  If so, the claim is somewhat lacking in persuasion. The reason for the question is that the trend at T=4 and T=5 has reversed and also the variance of the results in Figure 2 (a) is somewhat large.
3. In Table 2, ResNet56 selected as the student model is a larger-sized model than a few teacher models (e.g., vgg19). Is this the right KD experiment?
4. After the proposed KD process, is there no need for one more calibration of the student model? I know this paper focuses on increasing the accuracy of the student model, but since it has already been learned from a calibrated teacher model, I think it is also another contribution of this paper if there is no need to conduct the calibration process for student model. Simply put, I would like to know the ECE and ACE of the student model that applied the calibration of the student model in Table 7 after applying the proposed method.


For the rebuttal, if the weakness are well addressed this reviewer is willing to raise the review score.

After discussing the authors, my concerns are solved so that I have raised my score from 5 to 6.

---

> ### Author Response · Authors · 2023-11-16
>
> Dear reviewer,
>
> Thank you for recognizing our motivation on teacher selection matters on knowledge distillation and acknowledging better calibration errors can lead to a better teacher. We have thoughtfully examined your insightful recommendations and have formulated responses to the key points, which are provided below:
>
>
> **Weakness - It is necessary to assess whether the proposed method works well even if various calibration studies are applied. And also they should be covered in the related works.**
>
> Thank you for the suggestion. We provided its results in official comments, namely **Additional Experiments 1: KD results with various calibration methods**. Following your recommendation, we conducted experiments with various calibration methods and applied these methods to the teacher model in our KD setup. The result is in **Additional Experiment 1** in the official comment. We observed that KD performance improves with the application of diverse calibration methods.
>
> This experiment further supports our claim that applying calibration methods to the teacher enhances KD performance. Additionally, it shows that this improvement is not limited to a specific method but extends across various calibration approaches. We're grateful for your suggestion, as it has enriched our research.

---

> ### Author Response · Authors · 2023-11-16
>
> Below are our responses to the reviewer's questions.
>
> **Q1: choosing a student model in deployment condition.**
>
> In real-world deployments on edge devices, it's typical to select a student model that satisfies constraints such as memory bounds or inference time. Selecting a student that best learns from a given teacher is a challenging approach because we don’t know which student model is the best student for the teacher until we perform KD. Additionally, determining the most suitable teacher model for a student is also difficult, as indicated by previous studies[1][2], which have demonstrated that a teacher model's higher accuracy does not necessarily translate to higher accuracy for the student. Therefore, our research focused on identifying the criteria for a good teacher under the fixed constraints of edge devices, assuming a set student model.
>
>
> **Q2: Wouldn't it be possible to reverse when T=2 and T=4?**
>
> The mean and standard deviation presented in Figure 2 is derived from three experimental runs. There is a possibility of outliers, which can introduce fluctuations. However, if the experiments were to be repeated more times, it's likely that the standard deviation would decrease. While your concern about the apparent variance is valid, when considering the overall trend, it aligns well with our explanation.
>
> **Q3: Isn’t it ResNet56(student) is larger than some teacher model(e.g., VGG19)?**
>
> It seems there might be a common misconception regarding the size of ResNet56. According to the ResNet original paper by He et al.[3] that introduced the ResNet architecture, ResNet20, ResNet32, ResNet44, ResNet56, and ResNet110 are smaller architectures designed for the CIFAR-10 dataset, while ResNet18, ResNet34, ResNet50, and ResNet101 are larger architectures designed for ImageNet, thus having a larger parameter size. The parameter sizes for each architecture are as follows:
>
> | model | ResNet20 | ResNet32 | ResNet56 | ResNet110 | ResNet18 | ResNet34 | ResNet50 |
> | --- | :---: | :---: | :---: | :---: | :---: | :---: | :---: |
> | # params | 0.27M | 0.46M | 0.85M | 1.7M | 11.2M | 21.3M | 23.7M |
>
> In the aspect of parameters, ResNet56 has 0.85M parameters, while the parameter size of VGG19 is 20.1M. The smallest model we used as a teacher model is WRN-28-2, which has 1.5M parameters. Hence, the student model in our experiments is always smaller than the teacher model. Detailed parameter sizes of other teacher models are in Appendix A section of our paper.
>
>
> **Q4: After the proposed KD process, is there no need for one more calibration of the student model?**
>
> Thank you for suggesting a new insightful experiment. we performed an experiment to apply the calibration method to the student of Table 7, as you mentioned. We used temperature scaling for the calibration method. and for a fair comparison, we split the full-test set into the validation set and the test set with a 2:8 ratio. A validation set is used for calibration, and calibration metrics are reported with the test set. Then, the result is like below. Note that the detailed number differs from Table 7 due to the split test set, but it doesn’t change the tendency.
>
> |  |  | ShuffleNetV1 |  |  |
> | --- | --- | --- | --- | --- |
> | metric | ECE(%) | ECE_over(%) | ECE_under(%) | ACE(%) |
> | before calibration | **8.6057** | **8.3768** | 0.2289 | **0.1068** |
> | after calibration | 11.4309 | 11.3343 | **0.0966** | 0.1164 |
>
>
> |  |  | WRN-16-2 |  |  |
> | --- | --- | --- | --- | --- |
> | metric | ECE(%) | ECE_over(%) | ECE_under(%) | ACE(%) |
> | before calibration | **3.7519** | **3.0179** | 0.734 | **0.0936** |
> | after calibration | 10.5391 | 10.5186 | **0.0205** | 0.0976 |
>
> In this table, we can check calibration errors are better when we did not apply the calibration method. While asserting that no further calibration is needed after KD with a calibrated teacher would require more extensive experiments across a variety of models and calibration methods, our preliminary results suggest that using a calibrated teacher for KD results in a student model with robust calibration error. This finding can serve as a strong foundation for future research.
>
> We hope this comment successfully clarifies your concerns about our paper. We look forward to further discussions and appreciate your consideration.
>
>
> **References**
>
> [1] Jang Hyun Cho and Bharath Hariharan. On the efficacy of knowledge distillation. In Proceedings
> of the IEEE/CVF international conference on computer vision, pp. 4794–4802, 2019.
>
> [2] Seyed Iman Mirzadeh, Mehrdad Farajtabar, Ang Li, Nir Levine, Akihiro Matsukawa, and Hassan
> Ghasemzadeh. Improved knowledge distillation via teacher assistant. In Proceedings of the AAAI
> conference on artificial intelligence, volume 34, pp. 5191–5198, 2020.
>
> [3] He, K., Zhang, X., Ren, S., & Sun, J. (2016). Deep residual learning for image recognition. In Proceedings of the IEEE conference on computer vision and pattern recognition (pp. 770-778).

---

> > ### Comment · Reviewer_vJXm · 2023-11-17
> > **Thank you for good experiments.**
> >
> > Thank you for good experiments.
> > Response on the weakness and answer 4 (A4),
> > I think these experiments demonstrate the proposed method are valid.
> >
> > Response on the answer 1 (A1).
> > Albeit considering an edge device, I think it is still important to select a student model mimicking the best teacher model. The reason is there could be the different student models satisfying the constraint. Thus, I suggest to add this issue to the introduction and to emphasize what problem this paper aims to address.
> >
> > Response on the answer 2, 3 (A2, A3).
> > Thank you for your answer. I agree with you.
> >
> > The authors resolve my concerns. Thus, if there is no critical problem until rebuttal period, I will raise my score.

---

> > > ### Author Response · Authors · 2023-11-17
> > > **Thank You for Acknowledging Our Additional Experiments and Responses**
> > >
> > > Dear reviewer,
> > >
> > > Thank you for the quick response, and we are happy to our additional experiments and answers resolve your concerns.
> > >
> > > For additional response to Q1, we agree with the importance of selecting a better student model. It could be another research topic that explores which student models can learn well from a given teacher. Therefore, in the final version of our paper, we will clearly mention the scope of the problem that our research addresses.
> > >
> > > Thanks for the productive discussions and for acknowledging our research. Please feel free to raise any additional concerns or questions during the remainder of the discussion period.

---

### Official Review · Reviewer_f68K · 2023-10-31

**Soundness:** 2 fair
**Presentation:** 3 good
**Contribution:** 2 fair
**Rating:** 5
**Confidence:** 4

**Summary:**

This paper studies the knowledge distillation problem and reveals that there is a strong correlation between the calibration error of the teacher and the accuracy of the student. To reduce the teacher's calibration error, a temperature-based calibration is proposed.Extensive experiments demonstrate the effectiveness of the proposed method. With the help of MLLD, the proposed method achieves a new SOTA performance.

**Strengths:**

1. This paper is motivated well. The correlation between the calibration error of the teacher and the accuracy of the student is evaluated well (cf. Fig 1).
2. The experimental results are strong. This work establishes new SOTAs in this field.
3. This paper is written and organized well.

**Weaknesses:**

1. The proposed method is too simple. The temperature scaling has been proposed in the traditional KD. The technique contribution is small.
2. Althogh this work achieves SOTA performance, it is based on previous SOTA method MLLD. All experiments are conducted on the image classification task. Object detection or other tasks are not considered.

**Questions:**

None

---

> ### Author Response · Authors · 2023-11-16
>
> Dear reviewer,
>
> Thank you for recognizing the motivation of our research that reveals the relation between calibration error and KD performance. We have carefully reviewed your valuable suggestions, and accordingly, we have responded to the primary concerns as detailed below:
>
>
>
> **The proposed method is too simple, and temperature scaling has been proposed in the traditional KD:**
>
> The primary aim of our experiment utilizing temperature scaling was to demonstrate that reducing the teacher's calibration error can enhance KD performance.  Although temperature scaling itself is not our core contribution, our key contribution lies in demonstrating the effectiveness of reducing calibration error in boosting KD performance. The simplicity of our approach is advantageous, showing that even basic methods that reduce teacher calibration error can significantly improve KD. This finding suggests a foundation for future research to incorporate our core idea that lowering teacher calibration error can enhance KD methods even with more sophisticated methods. Because our research is based on a simple method, we think our results have the potential to influence the research field positively.
>
> Furthermore, **Additional Experiment 1** in the official comment extends our result beyond just temperature scaling. By applying our hypothesis to widely-used calibration methods, namely mixup and vector scaling, we show performance enhancement on KD. Which suggests that there exists room for enhancement. Also, considering vector scaling requires additional validation set and mixup also enhances teacher’s accuracy, we can see that our methods enable fair comparison between baseline and ours.
>
>
> **SOTA performance is based on the previous SOTA model:**
>
> Continuing from the above response, applying calibration methods to enhance the performance of SOTA models demonstrates that reducing calibration error can improve KD performance, even in complexly designed methods. This indicates that calibration errors could be considered in designing future SOTA methodologies. Our paper lays the groundwork for future KD method development, considering the teacher model's calibration error in method design could lead to a more effective KD method.
>
>
>
> **Experiment on Object detection:**
>
> Our method of reducing the calibration error for the teacher is very simple to apply to any classification problem. Therefore, we believe that our method would also work for other tasks involving classification as their sub-task. These include segmentation tasks and two-stage detection tasks such as Faster-RCNN, which employ classifiers in their final stages. In our final version, we would like to provide these results.
>
> We hope this comment successfully conveyed our contribution and addressed your concerns. We look forward to further discussions and appreciate your consideration.

---

> ### Author Response · Authors · 2023-11-20
> **Additional experiment with object detection**
>
> Thank you for suggesting an experiment on object detection. As you suggested, we conducted an additional experiment using our method for the object detection task. Since classifying objects is also included in object detection, and due to the simplicity of applying our method, we were able to easily apply our method to object detection by implementing temperature scaling on the teacher's classifier. The experiment results showed consistent performance improvements over the vanilla KD method. The results are as follows.
>
> |  | mAP | AP50 | AP75 | APl | APm | APs |
> | --- | --- | --- | --- | --- | --- | --- |
> | Teacher: ResNet101 | 42.04 | 62.48 | 45.88 | 54.60 | 45.55 | 25.22 |
> | Student: ResNet50 | 37.93 | 58.84 | 41.05 | 49.10 | 41.14 | 22.44 |
> ||||||||
> | KD | 38.35 | 59.41 | 41.71 | 49.48 | 41.80 | 22.73 |
> | KD+Ours | **39.04** | **60.74** | **42.21** | **50.38** | **42.35** | **22.88** |
>
> |  | mAP | AP50 | AP75 | APl | APm | APs |
> | --- | --- | --- | --- | --- | --- | --- |
> | Teacher: ResNet101 | 42.04 | 62.48 | 45.88 | 54.60 | 45.55 | 25.22 |
> | Student: ResNet18 | 33.26 | 53.61 | 35.26 | 43.16 | 35.68 | 18.96 |
> ||||||||
> | KD | 33.97 | 54.66 | 36.62 | 44.14 | 36.67 | 18.71 |
> | KD+Ours | **34.65** | **55.99** | **36.90** | **45.30** | **37.26** | **20.00** |
>
> |  | mAP | AP50 | AP75 | APl | APm | APs |
> | --- | --- | --- | --- | --- | --- | --- |
> | Teacher: ResNet50 | 37.93 | 58.84 | 41.05 | 49.10 | 41.14 | 22.44 |
> | Student: MobileNet-V2 | 29.47 | 48.87 | 30.90 | 38.86 | 30.77 | 16.33 |
> ||||||||
> | KD | 30.13 | 50.28 | 31.35 | 39.56 | 31.91 | 16.69 |
> | KD+Ours | **31.90** | **52.81** | **33.49** | **41.23** | **34.05** | **18.29** |
>
> The experiment was conducted following the commonly used object detection settings in previous knowledge distillation papers(Chen et al.[1], Zhao et al.[2], Jin et al.[3]). We conducted our experiment using the COCO2017 dataset and employed Faster RCNN, testing with Teacher-Student pairs of ResNet110-ResNet50, ResNet110-ResNet18, and ResNet50-MobileNetV2.
>
> Despite the simplicity of our method, the results show steady enhancements in performance. These results indicate that our main claim is not limited to classification tasks but is also applicable to object detection tasks. This additional experiment strongly supports our claim, so we will include it in the appendix of the final version. Thank you for suggesting this additional experiment. It has significantly strengthened the generality of our claim.
>
> We hope this additional experiment effectively addresses your concerns. Thank you for your insightful comments, and we look forward to further discussion.
>
> \
> [1] Chen, P., Liu, S., Zhao, H., & Jia, J. (2021). Distilling knowledge via knowledge review. In Proceedings of the IEEE/CVF Conference on Computer Vision and Pattern Recognition (pp. 5008-5017).
>
> [2] Zhao, B., Cui, Q., Song, R., Qiu, Y., & Liang, J. (2022). Decoupled knowledge distillation. In Proceedings of the IEEE/CVF Conference on computer vision and pattern recognition (pp. 11953-11962).
>
> [3] Jin, Y., Wang, J., & Lin, D. (2023). Multi-Level Logit Distillation. In Proceedings of the IEEE/CVF Conference on Computer Vision and Pattern Recognition (pp. 24276-24285).

---

### Official Review · Reviewer_F9hQ · 2023-10-31

**Soundness:** 3 good
**Presentation:** 3 good
**Contribution:** 3 good
**Rating:** 6
**Confidence:** 4

**Summary:**

Knowledge Distillation (KD) is a successful method for compressing deep learning models. However, recent research shows that a high-performance teacher doesn't guarantee a better student. This paper introduces a criterion for choosing an appropriate teacher: the teacher's calibration error, which correlates strongly with student accuracy. The paper also presents a temperature-based calibration method that reduces the teacher's error, leading to improved KD performance. This method can enhance other techniques and achieves a new state-of-the-art performance level when applied alongside current models.

**Strengths:**

- Practical Applicability: The introduced temperature-based calibration method offers a practical and effective solution for improving knowledge distillation performance.

- Clarity and Conciseness: The paper effectively conveys its key findings and contributions in a clear and concise manner, making it accessible to a wide audience.

- Comprehensive Experimental Validation: The paper backs its claims with thorough experimental evaluations conducted across multiple benchmarks and domains.

- Well-Structured Presentation: The paper is well-structured, with a clear introduction, detailed methodology, and comprehensive experimental results.

**Weaknesses:**

All in all, despite the simple idea and somewhat lack of novelty, the proposed method is technically sound and effective. It would be great if the authors of the paper can offer some sort of theoretical insights if possible.

One additional experiment that is worth conducting is: intuitively, in addition to ECE, the accuracy of the teacher model would also impact the quality of knowledge distillation. Does this mean that we should favor a teacher model with better ECE and accuracy? Are there any inherent tradeoffs between accuracy and ECE among different choices of teacher models? Is it always possible to get the ECE of different models to the same level by adjusting the temperature parameter? Personally I feel that it is worthwhile conducting additional experiments using the same student model, but different teacher models of different depth to demonstrate the effect of accuracy and ECE have on the quality of knowledge distillation. This would further strengthen the empirical contribution of the paper in my opinion.

Lastly, similar experiments were conducted previously to study the behavior between temperature and the effectiveness of knowledge distillation [1]. It would be great if a discussion is done in the related works section on the similarity and differences of this work.

[1] Zhang Z, Sabuncu M. Self-distillation as instance-specific label smoothing. Advances in Neural Information Processing Systems, 2020, 33: 2184-2195.

**Questions:**

It would be great if the authors of the paper can help address the question raised in the "weakness" section.

---

> ### Author Response · Authors · 2023-11-16
>
> Dear Reviewer,
>
> Thank you for acknowledging the practical applicability of our research, and thank you for your insightful feedback. We have carefully considered your constructive suggestions and addressed the main points as follows:
>
> **About Theoretical Insights and Novelty:**
>
> We provided detailed theoretical insights in Section 3.2. That is, 1) A teacher model with better calibration forms a more accurate probability distribution than one-hot labels, providing a mathematical foundation for computing the KL divergence loss in KD. 2) Temperature scaling enhances the label smoothing effect in KD, offering a more potent regularization effect. Our experiments in the paper support these insights.
>
> Addressing the concern about the perceived lack of novelty in our approach, we emphasized some key points. 1. While our method is simple and builds upon existing techniques, the meaningfulness of our work lies in its academic contribution rather than viewing it as a technical improvement in accuracy. 2. We shed light on the interplay between calibration and KD, which has not been deeply explored. 3. Our work significantly enhances the understanding of KD. We believe the primary contribution of our study lies in illustrating the profound influence that calibration has on the effectiveness of KD. This can be a robust groundwork for future explorations in KD and calibration methodologies.
>
> **Suggestions for an additional experiment and questions:**
>
> Thank you for the valuable suggestion on conducting experiments with various depths of teachers. We provided these results below. The table shows the results from experiments with different depths of ResNet and Wide-ResNet. Note that the numbers are from Tables 2 and 8 in our original paper. We believe this table will help answer several of your questions.
>
> | teacher | acc (%) | ACE (%) | student acc(wrn-16-2) | student acc(resnet56) |
> | :---: | :---: | :---: | :---: | :---: |
> | ResNet18 | 77.43 | 0.0938 | 75.75% | 74.36% |
> | ResNet34 | 78.41 | 0.0902 | 76.03% | 74.53% |
> | ResNet50 | **79.71** | **0.079** | **76.12%** | **75.13%** |
> | WRN-16-4 | 77.69 | **0.0833** | **76.25%** | **75.33%** |
> | WRN-28-4 | **78.56** | 0.0905 | 75.76% | 74.27% |
> | WRN-40-4 | 78.51 | 0.0914 | 75.59% | 75.00% |
>
> - **Q1. Do we favor a teacher model with better calibration and accuracy?:**
> In knowledge distillation literature, it has been repeatedly reported that teachers with better accuracy do not always result in a better-performing student [1][2].
> Our research begins by addressing the issue highlighted in those papers, namely the problem of higher accuracy teacher models not always producing superior student models.
> However, on the left side of Figure 1, we show there is a positive correlation between teacher’s accuracy and student accuracy. Furthermore, we find a stronger correlation between teacher’s ACE and student accuracy (right side of Figure 1). From these results, we can confirm that calibration error is a better criterion compared to accuracy. Using those two criteria, we can select a better teacher model compared to solely relying upon the teacher model’s accuracy. However, further investigation is needed to state this conclusively.
>
> - **Q2. Are there any inherent tradeoffs between accuracy and ECE among different choices of teacher models?:**
> To answer the question regarding the trade-off between accuracy and calibration error among different teacher models, our findings suggest that the relationship between calibration error and accuracy varies depending on the architecture. For instance, in the above Table, we observed that with ResNet, as the depth increases, both accuracy and calibration error improve. However, with Wide-ResNet, the worst accuracy was observed at depth 16, but it had the lowest ACE. Therefore, based on our experimental results, the trade-off between calibration error and accuracy cannot be straightforwardly established.
>
> - **Q3. Is it always possible to get the ECE of different models to the same level by adjusting the temperature parameter?:** Adjusting the temperature parameter to align the calibration error of different models to a similar level is challenging. While it's possible to set the temperature parameter using the validation set to achieve similar calibration errors across models, the calibration error in the test set can still vary despite these adjustments. What we did was reduce the calibration error by temperature scaling for a fixed teacher would produce a better student model in the KD.
>
> **References**
>
> [1] Jang Hyun Cho and Bharath Hariharan. On the efficacy of knowledge distillation. In Proceedings of the IEEE/CVF international conference on computer vision, pp. 4794–4802, 2019.
>
> [2] Seyed Iman Mirzadeh, Mehrdad Farajtabar, Ang Li, Nir Levine, Akihiro Matsukawa, and Hassan Ghasemzadeh. Improved knowledge distillation via teacher assistant. In Proceedings of the AAAI conference on artificial intelligence, volume 34, pp. 5191–5198, 2020.

---

> ### Author Response · Authors · 2023-11-16
>
> **Similarity and difference with “Self-distillation as instance-specific label smoothing”:**
> Thank you for pointing out the relevant paper. We were not aware of this work, and we appreciate your bringing it to our attention. This paper proposes that applying temperature scaling exclusively to the teacher during self-distillation results in improvements in performance and calibration. However, the experimental results of this paper are limited to self-distillation, while our research has established generality in the context of general knowledge distillation. Furthermore, we achieved results that calibration error can even be used as criteria for selecting teacher models. Because the experiment in this paper is well aligned with our research, adding it to the related work section of our final version could enhance the comprehensiveness of our paper.
>
> We hope these comments clarify our approach and address your concerns. We look forward to further discussions and appreciate your consideration.

---

### Author Response · Authors · 2023-11-16
**Additional Experiment 1: KD results with various calibration methods**

In our original paper, we propose that the application of the calibration method to teacher models enhances KD performance by using temperature scaling in Table 2. In addition to this experiment, we expanded our experiment to include various calibration methods. This additional experiment aimed to address that our contribution is not limited to specific calibration methods, but also can be applied with various calibration methods.

We conducted the experiment with various calibration methods and applied these methods to the teacher model at the top of Table 2 in our paper. We observed that KD performance improves with the application of diverse calibration methods. The results of these experiments are as follows:


## Additional Experiment 1: KD results with various calibration methods.
| teacher : | VGG19 | VGG13 | VGG16 | ResNet110 | WRN-28-2 | WRN-16-3 | WRN-40-2 | ResNet18 | WRN-16-4 |
| --- | :---: | :---: | :---: | :---: | :---: | :---: | :---: | :---: | :---: |
| teacher accuracy | 74.1% | 74.48% | 74.66% | 75.22% | 75.53% | 75.98% | 76.55% | 77.43% | 77.69% |
| mixup teacher accuracy | 75.83% | 76.64% | 76.49% | 76.38% | 76.86% | 76.42% | 78.22% | 79.94% | 78.19% |
| | | | | | | | | | |
| vanilla KD | 72.9% | 73.99% | 73.39% | 74.98% | 75.4% | 75.37% | 74.9% | 74.36% | 75.33% |
| **temperature scaling** | 73.37% | **74.33%** | **73.9%** | 75.39% | **76.12%** | **75.95%** | 75.2% | **74.68%** | 75.95% |
| **vectore scaling** | **73.47%** | 74.07% | 73.63% | **75.63%** | 75.73% | **75.95%** | **75.26%** | 74.48% | **76.15%** |
| **mixup** | **74.6%** | **74.4%** | **73.86%** | **76.08%** | **76.29%** | **76.65%** | **75.89%** | **74.84%** | **76.63%** |


We apply vector scaling and mixup. Vector scaling is a widely-used calibration method used in other calibration research like Guo et al. [1] and Nixon et al. [2]. Vector scaling has class size learnable parameters that normalize each output logit. So it has more complexity compared to temperature scaling. Mixup [3] is also a widely used augmentation technique that increases the model accuracy and effectively reduces calibration error [4][5]. Mixup is applied when training the teacher, and standard KD is performed by using a teacher trained with mixup. The results of our additional experiment consistently align with our finding that applying the calibration method to the teacher can enhance KD performance. This further substantiates our claim. Additionally, it shows that this improvement is not limited to a specific method but also can be applied across various calibration approaches. We are grateful for the reviewers' feedback, as this additional experiment enriched our research and more clearly supports our contributions. We adopted vector scaling and mixup as calibration methods since it is a widely used methods for calibration.

In our paper, we solely used temperature scaling, because vector scaling requires a separate validation set, which could potentially considered as unfair comparisons. In the case of mixup, while it does reduce calibration error, it also increases teacher accuracy, making it challenging to attribute the improvement in KD performance solely to reduced calibration error. Therefore, in our original paper, we chose temperature scaling for its fairness and its ability to reduce calibration error without affecting accuracy. Nevertheless, the experiment results of vector scaling and mixup support our claims and suggest the potential applicability of various calibration methods in KD.

**References**

[1]Guo, C., Pleiss, G., Sun, Y., & Weinberger, K. Q. (2017, July). On calibration of modern neural networks. In International conference on machine learning (pp. 1321-1330). PMLR.

[2] Nixon, J., Dusenberry, M. W., Zhang, L., Jerfel, G., & Tran, D. (2019, June). Measuring Calibration in Deep Learning. In CVPR workshops (Vol. 2, No. 7).

[3] Zhang, H., Cisse, M., Dauphin, Y. N., & Lopez-Paz, D. (2017). mixup: Beyond empirical risk minimization. arXiv preprint arXiv:1710.09412.

[4] Zhang, L., Deng, Z., Kawaguchi, K., & Zou, J. (2022, June). When and how mixup improves calibration. In International Conference on Machine Learning (pp. 26135-26160). PMLR.

[5] Thulasidasan, S., Chennupati, G., Bilmes, J. A., Bhattacharya, T., & Michalak, S. (2019). On mixup training: Improved calibration and predictive uncertainty for deep neural networks. Advances in Neural Information Processing Systems, 32.

---

### Meta-Review · Area_Chair_S3Zn · 2023-12-07

**Metareview:**

This paper studies an important problem in current Knowledge Distillation (KD) framework, i.e., what measures can indicate a good teacher in KD. This paper shows the correlation between the calibration error of teacher models and the accuracy of student models. The observation inspires teacher model selection by simply comparing calibration error and leveraging some existing calibration approaches to improve KD. The experiments show the effectiveness of the proposed calibrate-then-teach framework.

Despite the authors presenting some illustrative examples to analyze the correlation between calibration and distillation, I do not believe the current experiments sufficiently demonstrate their strong correlation. It is widely believed that current deep models face very severe miscalibration issue [1]. Therefore, I do not think comparing different teacher architectures in Figure 1, which plays an important role in this paper, are convincing given that all the models actually perform bad on calibration. A more convincing way to illustrate the correlation is to directly adjust the temperature of teacher models, and this experiment is presented in Figure 2. However, there are two severe weaknesses in Figure 2:

(1) Only two models are used. Since calibration error (ECE, ACE et al.) is a very sensitive metric, one must conduct experiments on a wide range of models to extract the correlation, like the experiments in Figure1. Even if the mentioned correlation does not exist, one can easily find two model architectures that show the same phenomenon in Figure 2, because it is intuitive that largely increasing or decreasing the temperature value will hurt both calibration and distillation.

 (2) Even with the two model architectures in Figure 2, it is not clear that the accuracy is strong correlated with calibration. In this experiment, the authors set the temperature interval as 0.5. However, since the calibration error is sensitive to temperature, for instance it is not clear that the best calibration is achieved with T=1.5 or some other values between 1.5 and 2 (calibration error changes significantly in this small range).

Besides these two major concerns, the reviewers also point out some other weaknesses such as lack of novelty (by reviewer F9hQ, f68K) and insufficient evaluation (by reviewer f68K). Based on the reviews, I am inclined to reject this paper.

**Justification For Why Not Higher Score:**

Please refer to metareview section.

**Justification For Why Not Lower Score:**

N/A.

---

### Decision · Program_Chairs · 2024-01-16

Reject